# Semi-supervised learning with a principled likelihood from a generative model of data curation

**Stoil Ganev and Laurence Aitchison**
Department of Computer Science
University of Bristol,
Bristol, UK
`laurence.aitchison@bristol.ac.uk`

## Abstract

We currently do not have an understanding of semi-supervised learning (SSL) objectives such as pseudo-labelling and entropy minimization as log-likelihoods, which precludes the development of e.g. Bayesian SSL. Here, we note that benchmark image datasets such as CIFAR-10 are carefully curated, and we formulate SSL objectives as a log-likelihood in a generative model of data curation. We show that SSL objectives, from entropy minimization and pseudo-labelling, to state-of-the-art techniques similar to FixMatch can be understood as lower-bounds on our principled log-likelihood. We are thus able to introduce a Bayesian extension of SSL, which gives considerable improvements over standard SSL in the setting of 40 labelled points on CIFAR-10, with performance of $92.2\pm0.3\%$ vs $88.6\%$ in the original FixMatch paper. Finally, our theory suggests that SSL is effective in part due to the statistical patterns induced by data curation. This provides an explanation of past results which show SSL performs better on clean datasets without any "out of distribution" examples. Confirming these results we find that SSL gave much larger performance improvements on curated than on uncurated data, using matched curated and uncurated datasets based on Galaxy Zoo 2.[1]

## 1 Introduction

To build high-performing deep learning models for industrial and medical applications, it is necessary to train on large human-labelled datasets. For instance, Imagenet (Deng et al., 2009), a classic benchmark dataset for object recognition, contains over 1 million labelled examples. Unfortunately, human labelling is often prohibitively expensive. In contrast obtaining unlabelled data is usually very straightforward. For instance, unlabelled image data can be obtained in almost unlimited volumes from the internet. Semi-supervised learning (SSL) attempts to leverage this unlabelled data to reduce the required number of human labels (Seeger, 2000; Zhu, 2005; Chapelle et al., 2006; Zhu & Goldberg, 2009; Van Engelen & Hoos, 2020). One family of SSL methods — those based on low-density separation — assume that decision boundaries lie in regions of low probability density, far from all labelled and unlabelled points. To achieve this, pre deep learning (DL) low-density separation SSL methods such as entropy minimization and pseudo-labelling (Grandvalet & Bengio, 2005; Lee, 2013) use objectives that repel decision boundaries away from unlabelled points by encouraging the network to make more certain predictions on those points. Entropy minimization (as the name suggests) minimizes the predictive entropy, whereas pseudo-labelling treats the currently most-probable label as a pseudo-label, and minimizes the cross entropy to that pseudo-label. More modern work uses the notion of consistency regularisation, which augments the unlabelled data (e.g. using translations and rotations), then encourages the neural network to produce similar outputs for different augmentations of the same underlying image (Sajjadi et al., 2016; Xie et al., 2019; Berthelot et al., 2019b; Sohn et al., 2020). Further developments of this line of work have resulted in many variants/combinations of these algorithms, from directly encouraging the smoothness of the classifier outputs around unlabelled datapoints (Miyato et al., 2018) to the "FixMatch" family of

---

[1]Our code: https://anonymous.4open.science/r/GZ_SSL-ED9E; MIT Licensed

algorithms (Berthelot et al., 2019b;a; Sohn et al., 2020), which combine pseudo-labelling and consistency regularisation by augmenting each image twice, and using one of the augmented images to provide a pseudo-label for the other augmentation.

However, some of the biggest successes of deep learning, from supervised learning to many generative models, have been built on a principled statistical framework as maximum (marginal) likelihood inference (e.g. the cross-entropy objective in supervised learning can be understood as the log-likelihood for a Categorical-softmax model of the class-label MacKay, 2003). Low-density separation SSL methods such as pseudo-labelling and entropy minimization are designed primarily to encourage the class-boundary to lie in low-density regions. Therefore they cannot be understood as log-likelihoods and cannot be combined with principled statistical methods such as Bayesian inference.

Here, we give a formal account of SSL methods based on low-density separation (Chapelle et al., 2006) as lower bounds on a principled log-likelihood. In particular, we consider pseudo-labelling (Lee, 2013), entropy minimization (Grandvalet & Bengio, 2005), and modern methods similar to FixMatch (Sohn et al., 2020). Thus, we introduce a Bayesian extension of SSL which gives $92.2 \pm 0.3\%$ accuracy, vs $88.6\%$ in the case of 40 labelled examples in the original FixMatch paper. We confirm the importance of data curation for SSL on real data from Galaxy Zoo 2 (also see Cozman et al., 2003; Oliver et al., 2018; Chen et al., 2020; Guo et al., 2020).

## 2 BACKGROUND

The intuition behind low-density separation objectives for semi-supervised learning is that decision boundaries should be in low-density regions away from both labelled and unlabelled data. As such, it is sensible to "repel" decision boundaries away from labelled and unlabelled datapoints and this can be achieved by making the classifier as certain as possible on those points. This happens automatically for labelled points as the standard supervised objective encourages the classifier to be as certain as possible about the true class label. But for unlabelled points we need a new objective that encourages certainty, and we focus on two approaches. First, and perhaps most direct is entropy minimization (Grandvalet & Bengio, 2005)

$$\mathcal{L}_{\text{neg entropy}}(X) = \sum_{y \in \mathcal{Y}} p_y(X) \log p_y(X) \tag{1}$$

where $X$ is the input, $y$ is the on particular label and $\mathcal{Y}$ is the set of possible labels. Here, we have followed the typical probabilistic approach in writing the negative entropy as an objective to be maximized. Alternatively, we could use pseudo-labelling, which takes the current classification, $y^*$, to be the true label, and maximizes the log-probability of that label (Lee, 2013),

$$\mathcal{L}_{\text{pseudo}}(X) = \log p_{y^*}(X) \qquad\qquad y^* = \arg\max_{y \in \mathcal{Y}} \log p_y(X). \tag{2}$$

Lee (2013) regarded pseudo-labelling as closely related to entropy miminization as the optimal value of both objectives is reached when all the probability mass is assigned to one class. However, they are not formulated as a principled log-likelihood, which gives rise to at least three problems. First, these methods cannot be combined with other principled statistical methods such as Bayesian inference. Second, it is unclear how to combine these objectives with standard supervised objectives, except by taking a weighted sum and doing hyperparameter optimization over the weight. Third, these objectives risk reinforcing any initial poor classifications and it is unclear whether this is desirable.

### 2.1 IN STANDARD SUPERVISED LEARNING, UNLABELLED POINTS SHOULD BE UNINFORMATIVE

It is important to note that under the standard supervised-learning generative model unlabelled points should not give any information about the weights. The typical supervised learning setup assumes that the joint probability factorises as,

$$P(X, \theta, Y) = P(X) P(\theta) P(Y|X, \theta), \tag{3}$$

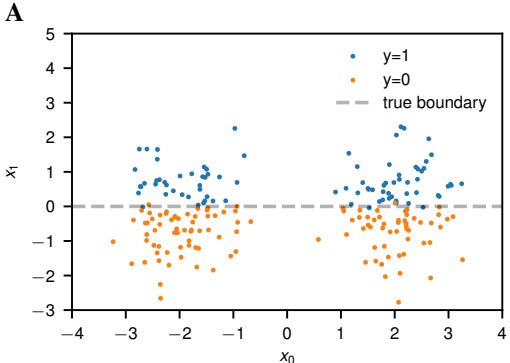 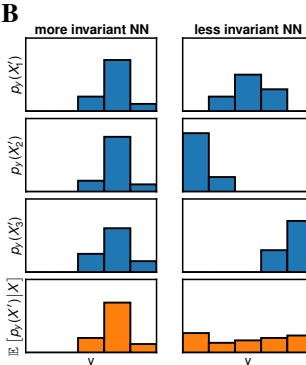

Figure 1: **A**. A toy dataset generated to illustrate the dangers of using the clustering of the input points to inform classification boundaries. The input features, $x_0$ and $x_1$ are plotted on the x and y-axes and the class is represented by colour. **B**. A schematic diagram demonstrating the effect of our principled likelihood incorporating data-augmentation on the certainty of predictions for different degrees of invariance. More invariant NNs (left) give similar predictive distributions for different augmentations (blue), and hence a certain averaged predictive distribution (bottom; orange). Less invariant NNs (right) give different predictive distributions for different augmentations (blue), and hence highly uncertain averaged predictive distributions (bottom; orange).

as the prior over weights, $\theta$ is usually chosen to be independent of the inputs (e.g. IID Gaussian). Thus, $X$ and $\theta$ are marginally independent and we cannot obtain any information about $\theta$ from $X$ alone. Formally, the posterior over $\theta$ conditioned on $X$ is equal to the prior,

$$\mathrm{P}\left(\theta|X\right) = \frac{\mathrm{P}\left(\theta, X\right)}{\mathrm{P}\left(X\right)} = \frac{\sum_{y \in \mathcal{Y}} \mathrm{P}\left(\theta, X, Y{=}y\right)}{\mathrm{P}\left(X\right)} = \frac{\mathrm{P}\left(\theta\right) \mathrm{P}\left(X\right) \sum_{y \in \mathcal{Y}} \mathrm{P}\left(Y{=}y|\theta, X\right)}{\mathrm{P}\left(X\right)} = \mathrm{P}\left(\theta\right).$$

as $1 = \sum_{y \in \mathcal{Y}} \mathrm{P}\left(Y{=}y|\theta, X\right)$. To confirm this result is intuitively sensible, note that there are many situations where encouraging the decision boundary to lie in low density regions would be very detrimental to performance. Consider a classifier with two input features: $x_0$ and $x_1$ (Fig. 1A). The class boundary lies in the high-density region crossing both clusters, so to obtain a reasonable result, the classifier should ignore the low-density region lying between the clusters. However, strong low-density separation SSL terms in the objective may align the cluster boundaries with the class boundaries, leading the classifier to wrongly believe that one cluster is entirely one class and the other cluster is entirely the other class. In contrast, supervised learning without SSL will ignore clustering and obtain a reasonable answer close to the grey dashed line. Importantly, this is just an illustrative example to demonstrate that without further assumptions, the standard supervised approach of ignoring unlabelled data is sensible; semi-supervised learning without loss of performance in such settings has been studied and is known as Safe SSL (Li & Zhou, 2014; Krijthe & Loog, 2014; Kawakita & Takeuchi, 2014; Loog, 2015; Krijthe & Loog, 2016; Guo & Li, 2018; Li et al., 2019).

## 2.2 A GENERATIVE MODEL OF DATA CURATION

Here, we give details on the generative model of data curation from Aitchison (2021). The key observation behind this model is that the dataset curators for image datasets like CIFAR-10 and ImageNet were very careful to include only images with a clear and unambiguous class label. For instance, for image datasets like CIFAR-10 and ImageNet, annotators were instructed that "It's worse to include one that shouldn't be included than to exclude one", and Krizhevsky (2009) "personally verified every label submitted by the annotators". In creating ImageNet, Deng et al. (2009) made sure that a number of Amazon Mechanical Turk annotators agreed upon the class before including an image in the dataset. Remarkably, this means that not all images have a valid class label. For instance, an image of a radio would not have a valid CIFAR-10 class-label, and hence it would have been rejected from the CIFAR-10 dataset. And further, because an "intelligent" decision has been made on whether or not to include the datapoint in the dataset, it might actually be possible to extract useful information from merely inclusion of the point in the dataset (which is precisely what SSL does).

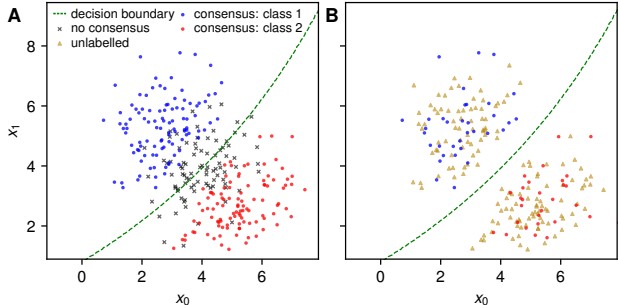

Figure 2: The generative model of data curation applied to a simple 2D dataset. Data from each class was sampled from a different Gaussian, and the true decision boundary (green dashed line) was given by the posterior probability of class given $(x_0, x_1)$. **A** Datapoints far from the decision boundary are unambiguous, so annotators agree and consensus is reached (red and blue points). Datapoints close to the decision boundary are ambiguous, so consensus is not reached (grey crosses). The consensus datapoints thus exhibit artificially induced low-density separation. **B** When using benchmark datasets such as CIFAR-10, the unlabelled points (yellow triangles) are selected from the consensus points (red or blue points) as the noconensus points are not available. The unlabelled points therefore also exhibit artificially induced low-density separation.

Formally, Aitchison (2021) considers a simplified generative model of consensus-formation between multiple annotators, as in Deng et al. (2009). In particular, for any image, $X$, $S$ human annotators, indexed $i$, are asked to give a label, $\{Y_i\}_{i=1}^S$ (e.g. using Mechanical Turk). Every annotator is forced to label every image and if the image is ambiguous they are instructed to give a random label. If all the annotators agree, $Y_1 = Y_2 = \cdots = Y_S$, they have consensus and the datapoint is included in the dataset. However, in the case of any disagreement, consensus is not reached and the datapoint is excluded. Concretely, the final label, $Y$ is $Y_1$ (which is the same as all the other labels) if consensus was reached and $\texttt{Undef}$ otherwise,

$$Y = \begin{cases} Y_1 & \text{if } Y_1 = Y_2 = \cdots = Y_S \\ \texttt{Undef} & \text{otherwise} \end{cases} \tag{4}$$

Taking $\mathcal{Y}$ to be the label set, we have $Y_i \in \mathcal{Y}$, and the final label, $Y$, could be any of the underlying labels in $\mathcal{Y}$, or $\texttt{Undef}$ if consensus is not reached, so $Y \in \mathcal{Y} \cup \{\texttt{Undef}\}$. When consensus was reached, the likelihood is,

$$\begin{aligned} \mathrm{P}\left(Y = y | X, \theta\right) = \mathrm{P}\left(\{Y_i = y\}_{i=1}^S | X, \theta\right) &= \textstyle\prod_{i=1}^S \mathrm{P}\left(Y_i = y | X, \theta\right) \\ &= \mathrm{P}\left(Y_i = y | X, \theta\right)^S = \left(p_y(X)\right)^S \end{aligned} \tag{5}$$

where we have assumed annotators are IID, and $p_y(X) = \mathrm{P}\left(Y_i = y | X, \theta\right)$ is the single-annotator probability.

## 3 METHODS

Now, we apply this model of data curation to SSL. The key question is what likelihood to use for unlabelled points. To understand this, remember that SSL methods are usually applied to benchmark datasets such as CIFAR-10 or ImageNet; these datasets were first carefully curated during the labelling process, implying that all points in the dataset have reached consensus. Critically, in typical benchmarks, unlabelled points are obtained by taking labelled points (which have reached consensus) and throwing away their labels (Fig. 2B). Under the generative model of consensus described above, we can obtain a probability of consensus by summing over class labels,

$$\mathrm{P}\left(Y \neq \texttt{Undef} | X, \theta\right) = \textstyle\sum_{y \in \mathcal{Y}} \left(p_y(X)\right)^S. \tag{6}$$

This probability is close to 1 (for $S > 1$) if the underlying distribution, $\left(p_y(X)\right)^S$ puts most of its mass onto one class, and the probability is smaller if the mass is spread out over classes. As such, the

likelihood "repels" decision boundaries away from unlabelled points, which is the common intuition behind low-density separation SSL methods, and which should be beneficial if class boundaries indeed lie in regions of low probability density away from both labelled and unlabelled points.

If noconsensus images are observed, we can include a likelihood term for those images,

$$\mathrm{P}\left(Y = \texttt{Undef}|X,\theta\right) = 1 - \mathrm{P}\left(Y \neq \texttt{Undef}|X,\theta\right) = 1 - \sum_{y\in\mathcal{Y}} \left(p_y(X)\right)^S. \tag{7}$$

If noconsensus images are not observed, we could in principle integrate over the underlying distribution over images, $\mathrm{P}\left(X\right)$. However, we do not even have samples from the underlying distributions over images (and if we did, we would have the noconsensus images so we could use Eq. 7). As such we omit this term, but the use of out-of-distribution (OOD) datasets as surrogate noconsensus points is an important direction for future work.

### 3.1 ENTROPY MINIMIZATION AND PSEUDO-LABELS ARE LOWER BOUNDS ON OUR PRINCIPLED LOG-LIKELIHOOD

Now, we try to understand whether our log-likelihood for noconsensus points bears any relation to previous successful methods for low-density separation based SSL. In particular, we begin by proving that entropy minimization forms a lower-bound on our log-likelihood (Eq. 6). First, we write the log-likelihood of consensus as an expectation over labels, $y$,

$$\log\mathrm{P}\left(Y\neq\texttt{Undef}|X,\theta\right) = \log\sum_{y\in\mathcal{Y}} p_y(X)\left(p_y(X)\right)^{S-1} = \log\mathbb{E}_{p_y(X)}\left[\left(p_y(X)\right)^{S-1}\right]. \tag{8}$$

Applying Jensen's inequality, the negative entropy gives a lower-bound on our log-likelihood,

$$\log\mathrm{P}\left(Y\neq\texttt{Undef}|X,\theta\right) \geq \mathbb{E}_{p_y(X)}\left[\log\left(p_y(X)\right)^{S-1}\right]$$
$$= (S-1)\sum_{y\in\mathcal{Y}} p_y(X)\log p_y(X) = (S-1)\mathcal{L}_{\text{neg entropy}}(X) \tag{9}$$

This bound is tight for a uniform predictive distribution,

$$\log\mathrm{P}\left(Y\neq\texttt{Undef}|X,\theta\right) = \log\sum_{y\in\mathcal{Y}}\left(p_y(X)\right)^S = \log\left(S\left(\tfrac{1}{S}\right)^S\right)$$
$$= -(S-1)\log S = (S-1)\mathcal{L}_{\text{neg entropy}}(X) \tag{10}$$

Pseudo-labelling forms an alternative lower bound on the log-likelihood which is obtained by noting that all $\left(p_y(X)\right)^S$ are positive, so selecting any subset of terms in the sum gives a lower bound,

$$\log\mathrm{P}\left(Y\neq\texttt{Undef}|X,\theta\right) = \log\sum_{y\in\mathcal{Y}}\left(p_y(X)\right)^S$$
$$\geq \log\left(p_{y^*}(X)\right)^S = S\log p_{y^*}(X) = S\mathcal{L}_{\text{pseudo}}(X). \tag{11}$$

The inequality holds if we choose $y^*$ to be any class, but will be tightest if we choose the highest probability class. This bound is tight for a predictive distribution that puts all its mass on $y^*$, so $p_{y^*}(X) = 1$ and $p_{y\neq y^*} = 0$

$$\log\mathrm{P}\left(Y\neq\texttt{Undef}|X,\theta\right) = \log\sum_{y\in\mathcal{Y}}\left(p_y(X)\right)^S = \log\left(p_{y^*}(X)\right)^S = S\log 1 = 0$$
$$S\mathcal{L}_{\text{pseudo}}(X) = S\log p_y^*(X) = S\log 1 = 0.$$

As such, entropy minimization and pseudo-labelling optimize different lower-bounds on our principled log-likelihood, $\log\mathrm{P}\left(Y\neq\texttt{Undef}|X,\theta\right)$, which gives a potential explanation for the effectiveness of pseudo-labelling and entropy minimization. Additionally, low-density separation SSL objectives encourages class-labels to be more certain. We can therefore expect pseudo-labelling to be the more relevant bound, as that bound is tight when the predictive distribution puts all its mass onto one class. In contrast, the entropy maximization bound is tight when the predictive distribution is uniform, which is discouraged by all low-density separation SSL objectives. This provides a potential explanation for the use of psuedo-labelling rather than entropy regularisation in modern SSL approaches such as (Sohn et al., 2020).

## 3.2 DATA AUGMENTATION PRIORS AND FIXMATCH FAMILY METHODS

The simple bounds above do not include data augmentation. However, modern FixMatch family methods combine data augmentation and pseudo-labelling. To understand FixMatch as a bound on a principled log-likelihood, we therefore need a principled account of data augmentation as a likelihood. Inspired by and Wenzel et al. (2020) (their Appendix K) and Nabarro et al. (2022) we consider a distribution, $\mathrm{P}\left(X'|X\right)$, over augmented images, $X'$, given the underlying unaugmented image, $X$. We choose the single-annotator predictive distribution as the average over predictive distributions for many different augmented images,

$$\mathrm{P}_{\mathrm{aug}}\left(Y_i{=}y|X,\theta\right) = \mathbb{E}\left[p_y(X')|X\right] \tag{12}$$

where $p_y(X')$ is the predictive probabilities resulting from applying the neural network to the augmented image, and remember $s \in \{1, \ldots, S\}$ indexes the annotator. To obtain a tractable objective in the supervised setting, Nabarro et al. (2022) recommend using a multi-sample version of Jensen's inequality, with $K$ augmented images denoted $X'_k$,

$$\log \mathrm{P}_{\mathrm{aug}}\left(Y_i{=}y|X,\theta\right) \geq \mathbb{E}\left[\log \tfrac{1}{K}\textstyle\sum_k p_y(X'_k)\big|X\right]. \tag{13}$$

Combining this single-annotator probability with our generative model of curation, we obtain,

$$\log \mathrm{P}_{\mathrm{aug}}\left(Y{=}y|X,\theta\right) = S \log \mathrm{P}\left(Y_s{=}y|X,\theta\right) = S \log \mathbb{E}\left[p_y(X')|X\right]$$
$$\geq S\, \mathbb{E}\left[\log \tfrac{1}{K}\textstyle\sum_k p_y(X'_k)\big|X\right], \tag{14}$$

The resulting objective for unlabelled points is,

$$\log \mathrm{P}_{\mathrm{aug}}\left(Y{\neq}\mathtt{Undef}|X,\theta\right) = \log \textstyle\sum_{y\in\mathcal{Y}} \mathrm{P}_{\mathrm{aug}}\left(Y{=}y|X,\theta\right) = \log \textstyle\sum_{y\in\mathcal{Y}} \mathbb{E}\left[p_y(X')|X\right]^S$$
$$\approx \log \textstyle\sum_{y\in\mathcal{Y}} \left(\tfrac{1}{K}\textstyle\sum_k p_y(X'_k)\right)^S, \tag{15}$$

where we approximate the expectation with $K$ different samples of $X'$, denoted $X'_k$. Unfortunately, this approach does not immediately form a bound on the log-likelihood due to the convex nonlinearity in taking the power of $S$. Nonetheless, one key problem with approximating machine learning losses is that the optimizer learns to exploit approximation errors to find a pathological solution that makes the objective unboundedly large. We appear to be safe from that pathology here, as we are simply forming predictions by averaging over $K$ augmentations of the underlying image. Nonetheless, to form a lower bound, we can follow FixMatch family algorithms by pseudo-labelling, i.e. by taking only one term in the sum for class $y^*$. FixMatch chooses $y^*$ by using the highest-probability class for a weakly-augmented image. An alternative approach is to choose the $y^*$ giving the tightest bound, i.e. $\arg\max_y \tfrac{1}{K}\sum_k p_y(X'_k)$. In either case,

$$\log \mathrm{P}_{\mathrm{aug}}\left(Y{\neq}\mathtt{Undef}|X,\theta\right) \geq \log \mathbb{E}\left[p_{y^*}(X')|X\right]^S \geq S\, \mathbb{E}\left[\log \tfrac{1}{K}\textstyle\sum_k p_{y^*}(X'_k)\big|X\right], \tag{16}$$

If $K = 1$ and $y^*$ is chosen using a separate "weak" augmentation, then this is exactly equal to the FixMatch objective for unlabelled points.

Note that both of these objectives (Eq. 15 and 16) promote reduced predictive uncertainty. Importantly, this does not just increase confidence in the single-augmentation predictive distributions, $p_y(X'_k)$, but also increases alignment between the predictive distributions for different augmentations (Fig. 1B). In particular, if the single-augmentation predictives are all highly confident, but place that high-confidence on different classes, then the multi-augmentation predictive formed by averaging will have low-confidence (Fig. 1B right). The only way for the multi-augmentation predictive to have high confidence is if the underlying single-augmentation predictive distributions have high confidence in the same class (Fig. 1B left), which encourages the underlying network to become more invariant. This makes sense: if data-augmentation changes the class predicted by the neural network, then any predictions *should* be low confidence. And it implies that combining principled data augmentation with a generative model of data curation automatically gives rise to an objective encouraging invariance.

## 3.3 A BAYESIAN EXTENSION OF SSL

Once we have derived a principled likelihood for SSL, $\mathrm{P}\left(Y|X,\theta\right)$, we can combine that likelihood with standard priors over Bayesian neural network weights, $\mathrm{P}\left(\theta\right)$, (in particular, we use independent Gaussians), and standard inference methods, specifically SGLD (Welling & Teh, 2011). This allows us to draw samples from the Bayesian posterior, $\mathrm{P}\left(\theta|X,\theta\right)$

$$\mathrm{P}\left(\theta|X,Y\right) \propto \mathrm{P}\left(Y|X,\theta\right)\mathrm{P}\left(\theta\right). \tag{17}$$

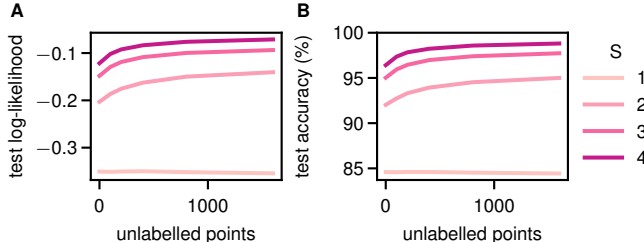

Figure 3: Test log-likelihood (A) and accuracy (B) for Langevin sampling for Bayesian SSL on toy datasets sampled from the model as a function of the number of unlabelled points. Note the lines for different values of $S$ are not directly comparable, as they use different data: each line uses data curated with a different value of $S$, and as we increase $S$ we curate more aggressively. More aggressive curation will improve performance as it eliminates ambiguous or difficult-to-classify points.

## 4 RESULTS

We began by giving a proof-of-principle for a Bayesian extension of SSL on a toy generated dataset, then we assessed the performance of Bayesian SSL on CIFAR-10. Finally, we confirmed the importance of curation for low-density separation based semi-supervised learning using Galaxy Zoo 2 as this was a real-world dataset which allowed us to generate matched curated and uncurated datasets.

### 4.1 BAYESIAN SSL ON A GENERATED DATASET

Our formulation of SSL as a likelihood implies that it should be possible to take entirely novel approaches, such as using low-density separation SSL in a Bayesian neural network (BNN). We considered a toy dataset generated from a "true" neural network model with one hidden layer and 30 hidden units, 5 dimensional inputs and 2 output classes. We did Bayesian inference in a network with the same architecture, with an independent Gaussian prior over the weights, with prior variance $1/$fan-in, which is necessary to ensure that the outputs have a sensible scale. We generated inputs IID from a Gaussian, then passed them through the "true" neural network, then sampled multiple categorical class-labels corresponding to different annotators. If all the simulated annotators agreed, consensus was reached and if any simulated annotators disagreed, consensus was not reached. We used 100 labelled datapoints, though not all of them will have reached consensus, and we used up to 1600 unlabelled points, though again not all of them will have reached consensus. Note that as the consensus/noconsensus status of a point arises from the generative model, we cannot independently specify the number of consensus/noconsensus points. We used Eq. (5) as the likelihood for labelled points, Eq. (6) as the likelihood for unlabelled points and Eq. (7) as the likelihood for noconsensus points. We sampled (and trained networks on) 500 datasets in parallel. We trained using Langevin dynamics with all data simultaneously (no minibatching) with no momentum and no rejection.

For a generative model with $S = 1$, consensus is always reached and the problem is equivalent to standard supervised learning. As such, we found no benefits from including unlabelled points for $S = 1$. In contrast, for any setting of $S > 1$ we found that increasing the number of unlabelled points improved the test log-likelihood (Fig. 3A) and the test accuracy (Fig. 3B).

### 4.2 BAYESIAN SSL ON CIFAR-10

Our work opens the possibility of combining Bayesian neural networks with SSL. We therefore designed a Bayesian variant of FixMatch, using the bound on the likelihood in Eq. (16), where we take $y^*$ from the weak augmentation and take the probabilities from the strong augmentation (remember the bound is valid for any choice of $y^*$). We used the same network, augmentation (RandAugment) and optimization schedule as in the original FixMatch paper, with the only modifications being to use 300 epochs rather than 1024 epochs, due to time constraints, and to add noise to the gradient updates. These noisy gradient updates convert SGD to a principled Bayesian inference method, SGLD (Welling & Teh, 2011). We used an isotropic Gaussian prior over the weights with variance $10^{-3}$ as that roughly corresponded to the usual $1/$fan-in variance initialisation, and we chose a temperature

Table 1: Accuracy of standard vs Bayesian FixMatch for different numbers of unlabelled points. Three seeds were used for all FixMatch and Bayesian FixMatch experiments except for FixMatch with 40 labelled points, where we used 12 seeds because the noise was much higher than that for any other setting. Bold indicates statistically significant improvements.

| number of labelled points: | 40 | 250 | 4000 |
|---|---|---|---|
| FixMatch | $88.5 \pm 0.7$ | $93.9 \pm 0.2$ | $94.6 \pm 0.1$ |
| Bayesian FixMatch | $\mathbf{92.2 \pm 0.3}$ | $93.9 \pm 0.1$ | $\mathbf{94.9 \pm 0.0}$ |
| FixMatch (unlab CIFAR-100) | $41.35$ | $62.63$ | $86.66$ |

of $T = 10^{-3}$ (Wenzel et al., 2020; Aitchison, 2021). The accuracy was computed by averaging over the last 100 epochs. As expected, we found that Bayesian FixMatch performed considerably better with the smallest number of unlabelled points (including better than the original FixMatch paper result of $88.6\%$), where uncertainty was highest, and gave similar performance otherwise (Tab. 1). Finally, we also tried training on on labelled points from CIFAR-10 and unlabelled points from CIFAR-100. As expected, we found a catastrophic drop in performance (Tab. 1).

### 4.3 GALAXY ZOO 2

Our data curation based likelihood predicts that low-density separation based SSL should be much more effective on curated than uncurated data. To test this prediction on real-world data, we turned to Galaxy Zoo 2 (GZ2) (Willett et al., 2013) which uses images from the Sloan Digital Sky Survey. This dataset is particularly useful for us as it has received only very minimal filtering based on criteria such as object brightness and spatial extent. We defined 9 labels by truncating the complex decision tree followed by the annotators (for further details see Aitchison, 2021). Further, as each GZ2 image has received $\sim 50$ labels, we can define a consensus coefficient by taking the fraction of annotators that agreed upon the highest probability class. We can then define a curated dataset by taking the images with consensus coefficient above some threshold within each class. Note that we needed to select images on a per-class basis, because annotators tend to be more confident on some classes than others, so taking the highest consensus coefficients overall would dramatically change the class balance. In particular, we used the top $8.2\%$ of images, which gave a full curated dataset of just over 20,000 images. Of those, we randomly selected 2000 as labelled examples, 10000 as test examples, and $0 - 6000$ as unlabelled examples. The images were preprocessed by center-cropping to $212 \times 212$ and then scaled to $32 \times 32$. We applied a FixMatch-inspired semi-supervised learning algorithm, with a standard supervised objective, with unlabelled objective given by Eq. (15) with $K = 2$ and $S = 10$. Data augmentation was given by vertical and horizontal flips, rotations from $-180°$ to $180°$, translations by up to 40% on both axes and scaling from 20% to 180%. We trained a ResNet18 with our maximum likelihood objective using SGD with a batch size of 500, a learning rate of 0.01 and 1500 epochs. We used an internal cluster of nVidia 1080 and 2080 GPUs, and the experiments took roughly 300 GPU hours.

We found that the test-log-likelihood for curated data improved slightly as more unlabelled points were included, whereas the test-log-likelihood for uncurated dramatically declined as unlabelled points were added (Fig. 4A–B). We saw strong improvements in test accuracy with the number of unlabelled points for curated datasets (Fig. 4C–D). Note that in Fig. 4C the error rate for curated datasets is already very small, so to see any effect we needed to plot the test error, normalized to the initial test error (Fig. 4D). For uncurated data, the inclusion of large numbers of unlabelled points dramatically worsened performance, though the inclusion of a small number of unlabelled points gave very small performance improvements (Fig. 4C–D). Thus, this experiment is consistent with the idea that the effectiveness of SSL arises at least in part from curation of the underlying dataset.

## 5 RELATED WORK

There are at least three main approaches to semi-supervised learning (Seeger, 2000; Zhu, 2005; Chapelle et al., 2006; Zhu & Goldberg, 2009). First there is low-density separation, where we assume that the class boundary lies in a region of low probability density away from both labelled

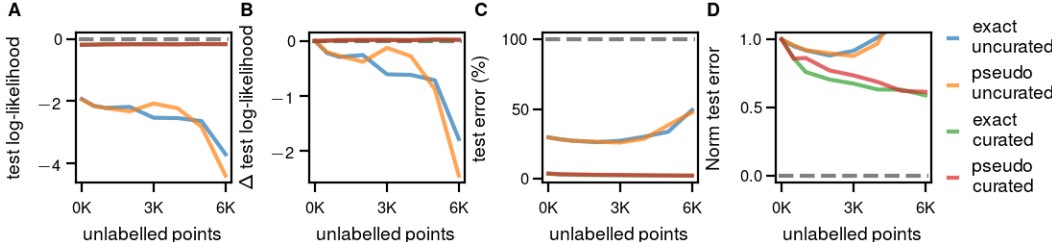

Figure 4: Test log-likelihood and error for curated and uncurated GZ2 datasets as a function of the number of unlabelled points. Exact corresponds to Eq. (15) (which is exact in the limit as $K \to \infty$) and pseudo corresponds to the pseudo labelling version of the augmented objective (Eq. 16). Note the green line (exact curated) is under the red line in A–C.

and unlabelled points. This approach dates back at least to transductive support vector machines (SVMs) where the model is to be tested on a finite number of known test locations (Vapnik, 1998; Chapelle et al., 1999). Those known test locations are treated as unlabelled points, and we find the decision boundary that perfectly classifies the limited number of labelled points, while at the same time being as far as possible from labelled and unlabelled data. Alternative approaches include pseudo-labelling and entropy minimization (Grandvalet & Bengio, 2005; Lee, 2013). Second, there are graph-based methods such as (Zhu & Ghahramani, 2002) which are very different from the methods considered here. Third, there are approaches that use unlabelled points to build a generative model of the *inputs* and leverage that model to improve classification (e.g. Kingma et al., 2014; Odena, 2016; Gordon & Hernández-Lobato, 2017). This approach was originally explored in a considerable body of classical work (e.g. McLachlan, 1975; Castelli & Cover, 1995; Druck et al., 2007) for a review, see Seeger (2000) and references therein. These approaches are fundamentally different from the SSL approaches considered here, as they require a generative model of inputs, while low-density separation methods do not. Generative modelling can be problematic as training a generative model can be more involved than training a discriminative model and because the even when the model can produce excellent samples, the high-level representation may be "entangled" (Higgins et al., 2017) in which case it may not offer benefits for classification.

## 6 DISCUSSION AND CONCLUSION

In conclusion, we showed that low-density separation SSL objectives can be understood as a lower-bound on a log-probability which arises from a principled generative model of data curation. This allows us to develop Bayesian SSL, which performs better in settings where there are a very small number of unlabelled points.

Our theory provides a theoretical understanding of past results showing that SSL is more effective when unlabelled data is drawn from the original, curated training set (Cozman et al., 2003; Oliver et al., 2018; Chen et al., 2020; Guo et al., 2020). In the extreme, our theory might be taken to imply that if data has not been curated, then SSL cannot work, and therefore that low-density separation SSL methods will not be effective in messy, uncurated real-world datasets. However, this is not the complete picture. Low-density separation SSL methods, including our log-likelihood, fundamentally exploit class-boundaries lying in low-density regions. As such, low-density separation could equally come from the real underlying data or could be artificially induced by data curation (Fig. 2). None of these methods are able to distinguish between these different underlying sources of low-density separation and as such any of them may work on uncurated data where the underlying distribution displays low-density separation. However, the possibility for curation to artificially induce low-density separation does imply that we should be cautious about overinterpreting spectacular results for low-density separation based SSL obtained on very carefully curated benchmark datasets such as CIFAR-10.

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
