# OpenReview forum: "Semi-supervised learning with a principled likelihood from a generative model of data curation"
_ICLR.cc/2023/Conference — ICLR 2023 poster_

### Official Review · Reviewer_fBnW · 2022-10-19

**Confidence:** 4
**Correctness:** 4
**Technical Novelty And Significance:** 2
**Empirical Novelty And Significance:** 3
**Recommendation:** 5

**Clarity, Quality, Novelty And Reproducibility:**

- Sec 2.2 uses `n` annotators and switches to `S` for the remainder of the paper
- (10) contains an error. It should be $\log(S (1/S)^S) = (1-S)\log S$, i.e., you are missing a negative sign. Similarly, given that the derivation is about lower terms to a likelihood that is to be maximized (9,10) would read better if they were formulated in terms of an entropy objective that is to be maximized instead of the current negative entropy, giving then $(1-S)L_\text{entr}$ for both. (I.e., as in (11) which is explicitly formulated as a maximization task)

**Clarity and Quality:** The paper itself is well-written but contains an error in the derivation

**Novelty:** The approach heavily builds on the prior work by Aitchison (2021) and is a rather minor extension of it.

**Reproducibility:** In its current form reproducibility is not given for most of the experiments.



## Specific questions
- Q1: Fig 2 speaks of _our generative model_. Can the authors clarify that part, i.e., to which degree does it differ from the example and model of Aitchison (2021) Figure 3?
- Q2: Fig 3. Can the authors comment on why the test log-likelihood seems to improve with more samplers (A) but the test accuracy decreases? Similarly, the caption lacks an explanation wrt. what the change in B, D consists of which has to be guessed by the reader.
- Q3: As stated above, the proposed approach could be used in a deterministic setting as well. Can the authors comment on how a maximum likelihood approach would perform in Sec 4.1 and 4.2 and vice versa how a BNN would perform in Sec 4.3 (e.g., Aitchison (2021) relies on an SGLD approach in that experiment)?

## Minor comments
- The method is restricted to the low-density approach of semi-supervised learning. But this restriction is acknowledged in Sec 5 and properly defended.
- Sec 2
- The experiments as summarized in Fig 3/4 lack standard deviations

## Typos
- Inconsistency in the equation typography, i.e., some end with `.`,`,` others don't
- End of Sec 3.1: _psuedo-labeling_

**Strength And Weaknesses:**

## Strengths
The paper is well argued for and shows clear results upon the baseline the authors are considering and demonstrates in varied ways their assumptions

## Weaknesses
- The strong emphasis on this being _Bayesian SSL_ seems to me to be too strong. While the proposed approach gives us a proper likelihood the method and approach lack any intrinsic "Bayesianity". Dropping the Bayes terms the paper could have just as well been formulated as a novel likelihood for a maximum likelihood approach. The authors themselves 'simply' use it in this function in their third experiment.
- Inconsistent notation and minor errors (see below)
- A lot of the hyperparameters and experimental details are missing, or partially available but then hidden in the provided code. E.g., Sec 4.1 mentions an architecture for creating the toy data set but lacks any information on whether the trained networks have the same architecture. Sec 4.3 lacks a complete discussion on how the authors arrived at their final nine labels, making replicability and reproducibility impossible. If the preprocessing follows Aitchison (2021) then please properly cite and discuss this fact.

**Summary Of The Paper:**

The authors consider the fact that our common benchmark data sets are curated, in the sense of their data points being preselected by the consensus of a set of labellers. Aitchison (2021) already noted this and designed a proper likelihood that takes account of this fact. The authors build upon his work by moving into the semi-supervised setting where they show how to formulate two common approaches (entropy minimization, pseudo-labelling) as lower bounds of this likelihood. The approach is then evaluated in several experiments ranging from synthetic to real-world.

**Summary Of The Review:**

While the paper contains some interesting formulations and results that are well worth further investigation in its current form it is rather incremental and contains several technical flaws (derivation and esp. wrt reproducibility) that need to be addressed.

---

> ### Author Response · Authors · 2022-11-07
> **Response**
>
> > The strong emphasis on this being Bayesian SSL seems to me to be too strong. While the proposed approach gives us a proper likelihood the method and approach lack any intrinsic "Bayesianity". Dropping the Bayes terms the paper could have just as well been formulated as a novel likelihood for a maximum likelihood approach. The authors themselves 'simply' use it in this function in their third experiment.
>
> Agreed absolutely (the title is about the likelihood rather than about Bayes). But a key part of the reason we care about a principled likelihood is so that we can do principled Bayesian inference, and this what two of the three experiments are about.
>
> > E.g., Sec 4.1 mentions an architecture for creating the toy data set but lacks any information on whether the trained networks have the same architecture.
>
> Yes, we train and test on the same architecture, and have added a note on this.
>
> > Sec 4.3 lacks a complete discussion on how the authors arrived at their final nine labels, making replicability and reproducibility impossible. If the preprocessing follows Aitchison (2021) then please properly cite and discuss this fact.
>
> All details are available in Appendix C of Aitchison (2021); we have added a note about this.  We have also added details e.g. of priors and prior variances.
>
> > Sec 2.2 uses n annotators and switches to S for the remainder of the paper
>
> Fixed (changed everything to $S$).
>
> > (10) contains an error.
>
> Fixed.  Note the error was isolated to that line: we really are maximizing the negative entropy (i.e. minimizing the entropy), as is standard in low-density separation based SSL (Grandvalet & Bengio 2005).
>
> > Q1: Fig 2 speaks of our generative model. Can the authors clarify that part, i.e., to which degree does it differ from the example and model of Aitchison (2021) Figure 3?
>
> We have switched "our" -> "the".  Our part of the generative model is the insight that unlabelled points in SSL have reached consensus, and therefore can be assigned an informative likelihood by summing over $Y \in \mathcal{Y}$.
>
> > Q2: Fig 3. Can the authors comment on why the test log-likelihood seems to improve with more samplers (A) but the test accuracy decreases?
>
> The lines for different values of S are not directly comparable, as they use different data. Specifically, each line uses data curated with that value for S, and as we increase S we curate more aggressively. Further, more aggressive curation will improve performance as it eliminates ambiguous or difficult-to-classify points. We have added a discussion of this point to the caption.
>
> > Similarly, the caption lacks an explanation wrt. what the change in B, D consists of which has to be guessed by the reader.
>
> Fixed.  B and D show the change in test log-likelihood and accuracy wrt to the initial value with zero unlabelled points.
>
> > Q3: As stated above, the proposed approach could be used in a deterministic setting as well. Can the authors comment on how a maximum likelihood approach would perform in Sec 4.1 and 4.2 and vice versa how a BNN would perform in Sec 4.3 (e.g., Aitchison (2021) relies on an SGLD approach in that experiment)?
>
> * 4.1: A determinstic approach is likely to work well here, but potentially less well than a full Bayesian approach.
> * 4.2: We believe that stochasticity is critical here to obtaining performance improvements over already well-tuned existing methods.
> * 4.3: We believe that Bayes would give a bit more performance, but would not give any dramatic changes.

---

### Official Review · Reviewer_JPM3 · 2022-10-25

**Confidence:** 4
**Correctness:** 4
**Technical Novelty And Significance:** 3
**Empirical Novelty And Significance:** 4
**Recommendation:** 8

**Clarity, Quality, Novelty And Reproducibility:**


## Novelty

I think there's plenty of contributions here:

1) the observation that most existing SSL benchmarks are tightly curated (e.g. require multi-annotator agreement) is not widely known/mentioned or used within approaches, but this paper makes a clear case that it should be better understood.

2) the interpretation of negative entropy and pseudolabel loss functions as likelihoods appears new to me (despite these ideas being popular for a decade plus) and it nicely offers an explanation of why pseudolabel is more in favor now (its bound is tighter).

3) building a "Bayesian" FixMatch that outperforms standard FixMatch at small datasets sizes (Tab 1) is nice to see


## Quality

### Why does accuracy get worse as S increases in Fig 3?

Fig 3C clearly shows that S=4 annotators (dark purple) reach 70-73% accuracy, while S=1 reaches ~85% accuracy. Can the authors please explain?

As the number of annotators S increases, I would expect that on *labeled* test data the task of classification becomes easier, as only examples where the true class distribution gave many samples of the same class are included. Requiring more consensus should discard examples near decision boundaries, and thus make learning sharp boundaries easier.

I guess the test dataset itself is changing as S is changing? Is the ratio of pos/neg examples changing? The size of test set changing?
I think would be useful to clarify this in the figure/caption.

Because panels B and D are somewhat redundant re-visualizations of the same information in panels A and C, I wonder if the figure would be better off only showing A and C, while also showing some other visualization that makes the nature of the task a bit more clean (e.g. show a scatterplot in 2d of the examples, colored by class label, somehow indicating which examples are "consensus" or not under each of the S values)

### Would results improve if used exact likelihood instead of the bound?

Bayesian FixMatch uses Eq 16 as the likelihood in CIFAR-10 experiments.
In earlier toy experiments, Eq 6 is used as likelihood, and Eq 16 is only a lower bound of Eq 6.

Could we use Eq 6 (the ideal likelihood, not a bound) in CIFAR-10 and get better results than Bayesian FixMatch in Table 1?
My guess is the augmentation hiding inside FixMatch is delivering generalization gains that wouldn't be possible with Eq 6, even though it is the "ideal" likelihood, but the explanation in this paper for why the Bayesian approach is better does not seem to account for that.


### Unclear what prior is used in the Bayesian FixMatch

Naturally, Bayesian methods require a prior and likelihood. I don't see a clear description of the prior used in Sec. 4.2. Can the authors please elaborate?

Probably it is just an independent Gaussian on each weight, but really should be specified in the main paper.

### Use of a temperature in the Bayesian FixMatch result needs more explanation

The use of the temperature (as in cold posterior paper by Wenzel et al. 2020) to me spoils the interpretation as truly "Bayesian"... I'd like to see some analysis of how necessary this is (what is performance with temperature 1.0?)

### Clarify purpose of training on CIFAR-100 images in Table 1?

I'm not sure why the authors report the third line in Table 1, showing FixMatch (a baseline) trained on unlabeled data from a different dataset (CIFAR-100).

Naturally, it is reassuring to know that using different unlabeled data does worse, as we might expect, but it's not clear how this experiment lends credibility to the presented Bayesian approach or the overall interpretation of this paper that a model of "curation" is useful to SSL.


## Clarity

Overall I found the manuscript's high-level points came through, but it was unfortunately tough to read in places, with awkward/abrupt transitions and frequent notation switching.

I'd recommend that transitions between subsections get revised to help the reader keep a narrative thread of where this is all going. For example, Sec. 2.2. immediately dives into a generative model of data curation without explaining how it fits in, and Sec. 3.1 dives into a proof of a bound without motivating why it is useful.

Here's a few further problematic clarity issues:

* Sec 2 never defines the symbols X or Y (easy enough to guess what they mean)
* Symbol Y_sup in Equation below Eq 2 is never defined, not clear if different from Y without subscript
* Sec. 2 uses "n" to denote number of annotations, this switches to "S" in Sec. 3 without explanation. I'd suggest keeping S, since "n" often denotes training data size.


**Strength And Weaknesses:**

# Strengths

* Highlights that curation (multi-annotator agreement) is behind many popular SSL benchmarks, which is less widely understood/used than it should be
* Principled approach to defining likelihoods that correspond to bounds on popular SSL objectives
* Showing how to use these likelihoods within a Bayesian model seems valuable
* Analysis in Sec 2.1 and Experiments on toy data are convincing: if data comes from a single-annotator model, hopes for SSL are dim (any unlabeled data is uninformative!)

# Weaknesses

* Why does accuracy get worse as S increases in Fig 3 on toy data?
* Use of a bound (not exact likelihood) in CIFAR experiments may hide role of augmentation
* Unclear what prior is used in the Bayesian FixMatch
* Use of a temperature in the Bayesian FixMatch result needs more explanation
* Writing needs some work: Lots of abrupt transitions and confusing notation

These weaknesses are elaborated further below (under Quality/Clarity)

**Summary Of The Paper:**

The paper looks at the topic of semi-supervised learning for classification, through a probabilistic lens. The starting point is the observation that many common SSL datasets (e.g. CIFAR) are assembled via multi-annotator agreement protocol (images are only included if many humans agree on their class label). This leads to a multi-annotator consensus generative model (previously described in Aitchison 2021). Eq. 6 gives this model's likelihood for an image to be a *consensus* image (e.g. images that would be included even in unlabeled set because many annotators agree).

The new thing in this paper is the realization that this likelihood can be lower bounded by many common SSL objectives, included entropy (Eq 9) or pseudolabel (Eq 11) or FixMatch (Eq 16). This gives a nice interpretation of what these objectives are doing (maximizing the likelihood of a model that is aware that consensus is needed).

Experiments on toy data demonstrate properties of this model, including:

1) Fig 3 suggests that if S =1 (multiple annotators are not needed), unlabeled data does not help under this model, while if S > 1 (several annotators need to agree), unlabeled data can help

2) Tab 1 shows that a Bayesian version of FixMatch, using their view of FixMatch's loss as a likelihood, does better than FixMatch on CIFAR-10 with few labeled images

3) Fig 4 shows that on Galaxy Zoo data, datasets that are *curated* show gains from unlabeled data while *uncurated* sets show losses not gains in classifier performance as more unlabeled data are added



**Summary Of The Review:**

Overall I like the core ideas here: the observation that curation is behind many standard SSL datasets is potentially powerful though underused currently: the methods here suggest that multi-annotator curation can be exploited (if present) via SSL but that datasets with only single annotator curation may not be usefully pursued with SSL methods.

<strike>
However, there's too many writing clarity issues and overall puzzling results to give this a high score currently. I hope discussion period clarifies things, I'd be willing to raise my score given a satisfactory response.
</strike>

**Update after discussion**: Clarity issues have largely been resolved, and the past improvements suggest remaining ones will also be resolved in good faith. Given the significance of the contributions here, I would vote for and argue for acceptance.

---

> ### Author Response · Authors · 2022-11-07
> **Response**
>
> > Why does accuracy get worse as S increases in Fig 3 on toy data?
>
> The lines for different values of S are not directly comparable, as they use different data. Specifically, each line uses data curated with that value for S, and as we increase S we curate more aggressively. Further, more aggressive curation will improve performance as it eliminates ambiguous or difficult-to-classify points. We have added a discussion of this point to the caption.
>
> > Unclear what prior is used in the Bayesian FixMatch
>
> We used an isotropic Gaussian prior with variance $10^{-3}$, and have added a note about this.
>
> > The use of the temperature (as in cold posterior paper by Wenzel et al. 2020) to me spoils the interpretation as truly "Bayesian"... I'd like to see some analysis of how necessary this is (what is performance with temperature 1.0?)
>
> We agree, this is annoying, and performance does degrade as temperature increases.  We speculate that it could be due to FixMatch hyper parameters being tuned for the non-Bayesian setting (we do not have the compute resources necessary to repeat the extensive hyperparameter searches from the original FixMatch paper).
>
> > Would results improve if used exact likelihood instead of the bound?
>
> Eq. 6 is not the ideal likelihood, and Eq. 16 is not an approximation to/bound on Eq. 6. Instead, Eq. 6 and Eq. 16 are likelihoods (or bound on likelihoods) for different generative models. Eq. 6 gives the likelihood for a generative model without data-augmentation. In contrast, Eq. 16 gives the likelihood for a generative model with data-augmentation (Eq. 16).  See Nabarro et al. (2022) for more details on principled Bayesian models that incorporate data augmentation in the purely supervised setting.  The key contribution here is that even sophisticated FixMatch like schemes that use data augmentation can be brought into the Bayesian framework in a principled fashion.
>
> > Clarify purpose of training on CIFAR-100 images in Table 1?
>
> Agreed.  This is just a sensible sanity check.
>
> > I'd recommend that transitions between subsections get revised to help the reader keep a narrative thread of where this is all going. For example, Sec. 2.2. immediately dives into a generative model of data curation without explaining how it fits in, and Sec. 3.1 dives into a proof of a bound without motivating why it is useful.
>
> We have updated the transitions for Sec. 2.2, Sec. 3 and Sec. 3.3.
>
> > Sec 2 never defines the symbols X or Y (easy enough to guess what they mean)
>
> Fixed
>
> > Symbol Y_sup in Equation below Eq 2 is never defined, not clear if different from Y without subscript
>
> Fixed (deleted sup).
>
> > Sec. 2 uses "n" to denote number of annotations, this switches to "S" in Sec. 3 without explanation. I'd suggest keeping S, since "n" often denotes training data size.

---

> > ### Comment · Reviewer_JPM3 · 2022-11-19
> > **Thanks for the response! Many things have been cleared up.**
> >
> > Thanks to the authors for a thoughtful response. I'm really sorry I'm so late to reply, but here's my thoughts...
> >
> > First, I can mark as fully resolved the request for the prior and the query about temperature.
> >
> > Two other points I have a bit more to say about...
> >
> > ### RE why does accuracy in Fig 3 get worse as S increases
> >
> > I agree with what the authors write, that it *should* be the case that
> >
> > > more aggressive curation will improve performance
> >
> > However, that's not exactly what we see in Fig 3.... S=4 is more aggressive curation (4 different annotators need to agree). Yet in Fig 3 C we see accuracy *decrease* for larger S: S=4 (dark purple) has 70% accuracy, while S=1 (lightest color line, just one annotator) has >80% acc
> >
> > ### RE the difference between Eq 6 and Eq 16
> >
> > OK. I've read again carefully, I agree that with improved understanding my original question about these equations doesn't really make sense. I agree the overall contribution of Eq 16 (showing how FixMatch can be brought into the Bayesian framework) is nice.
> >
> > I do think my confusion was reasonable, and speaks to a bit of the clarity issues I had raised. I guess my confusion stems from how the two are written, which if I copy the left-hand-side of each and look at them side by side, I get:
> >
> > \begin{align}
> > P( Y \neq undef | X, \theta) &= \ldots \quad (6)
> > \\\\
> > \log P( Y \neq undef | X, \theta ) &\geq \ldots \quad (16)
> > \end{align}
> >
> > I think a reasonable reader could expect to conclude that 16 is a lower bound on the log of Eq 6. If they really are two *different* models, probably marking them as such (maybe use subscript P_A for the bottom case, for augmentation) is necessary to avoid confusion.
> >
> > I hope the authors consider revising a bit futher, just to prevent future confusion.
> >
> > ## Overall score improved to accept
> >
> > Now that many clarity issues have been resolved, I'd vote for acceptance. I like the overall thrust of this work.
> >
> > I do hope the authors revise to improve the discussion of why accuracy in Fig 3 decreases with curation (larger S), but I don't see this as show-stopping.

---

> > > ### Author Response · Authors · 2022-11-19
> > > **Response**
> > >
> > > Thanks!
> > >
> > > We have added an $P_{\rm aug}$ label to the probability distribution in the augmented model, to clear up that confusion.
> > >
> > > Good catch!  If you look at the current Fig. 3, log-likelihoods get better with larger S, while accuracies get worse.  This is very odd, and points to a bug in the evaluation.  Indeed, we did find a bug in the calculation for the accuracy.  Fixing that bug, the accuracy behaves very similiarly to the log-likelihood.  In particular, both the log-likelihood and accuracy increase with S, as expected.
> > >
> > > We have a pdf with these changes ready to go, but the "Revise submission" button seems to have disappeared.

---

### Official Review · Reviewer_3m8s · 2022-10-25

**Confidence:** 4
**Correctness:** 4
**Technical Novelty And Significance:** 2
**Empirical Novelty And Significance:** 2
**Recommendation:** 3

**Clarity, Quality, Novelty And Reproducibility:**

I feel that the paper has some reproducibility issue, since not all the detail implementations are given for the experiments. For example, what is S value for the Galaxy zoo experiment; page 7, it says "We used Eq. (5) as the likelihood for labelled points, Eq. (6) as the
likelihood for unlabelled points and Eq. (7) as the likelihood for noconsensus points ",  I have no idea what is the likelihood for no-consensus unlabelled points.

**Strength And Weaknesses:**

Strength: The likelihood for curated/uncurated data set seems interesting, and it successfully establishes a relationship to entropy minimization and  pseudo-labelling.

Weakness: The construction of likelihood is too artificial, aiming to appeal low density separation idea. How to choose S? How to determine a labelled/unlabelled observation is curated or not, beyond common data sets such as CIFAR10? Unlike penalty term which are artificial designed to incorporate prior knowledge, likelihood should reflect data generation rule and tuning-free.

One experiment shows the improvement when unlabelled curated data is added, and the performance drop with more uncurated data. But this is not a convincing justification of the main of idea of the paper. The theory shows that log likelihood is lower bounded by the entropy or pseudo labelling loss under the generative model of curated data. It does not say what happens when the data is uncurated. To establish empirical justification, one also needs to show that the performance will increase with added uncurated data if likelihood (7) is used.

The paper claims it is a Bayesian approach, but it never mention anything about prior setting. For all the results displayed, are they based on posterior MAP or posterior mean?

Figure 3, even when there is no unlabelled data, the test accuracy is different w.r.t. the choice of S. S=0 yields the best test accuracy. Please comment on this

What is the test likelihood (Figure 3/4)?


**Summary Of The Paper:**

The paper aims to formulate the low-density separation term in SSL as a log-likelihood term from a generation model of data curation. Based on which, a Bayesian modeling is possible, and some new insight on why SSL works for curated data set.

**Summary Of The Review:**

Overall, I feel the paper is more like playing with a math trick, but can not deliver more insight. The low-density separation SSL assumes the decision boundary is not close to any data point, thus, even without any math, it implies that all data, including unlabelled data are away from decision boundary, i.e., they must be curated data. A mathematical formulate should provide us with more subtle insight of this matter, and this paper seems fail to do it.

---

> ### Author Response · Authors · 2022-11-07
> **Response (2/2)**
>
> > For example, what is S value for the Galaxy zoo experiment
>
> S=10, we have added a note about this.
>
> > "We used Eq. (5) as the likelihood for labelled points, Eq. (6) as the likelihood for unlabelled points and Eq. (7) as the likelihood for noconsensus points ", I have no idea what is the likelihood for no-consensus unlabelled points.
>
> I'm not quite sure what the question is here (i.e. whether its an issue about labelled vs unlabelled no-consensus points or no-consensus points in general). Anyway, during the curation process, $S$ annotators assign a label to a point. If they all agree, then consensus is reached, and the point is given a label.  If they disagree, then this is a no-consensus point, and it does not make sense to apply a label.  Importantly, there is thus no labelled vs unlabelled distinction for no-consensus points. All no-consensus points are unlabelled, because these are precisely points where the annotators could not reach consensus on the label.  Now, we can ask about the probability that the annotators will fail to reach consensus.  That's the likelihood for noconsensus points, given by Eq. 7.

---

> ### Author Response · Authors · 2022-11-07
> **Response (1/2)**
>
> > The construction of likelihood is too artificial, aiming to appeal low density separation idea.
>
> The likelihood was not constructed to appeal to the low-density separation idea: that's the surprising thing!  The likelihood was designed to capture the cold posterior effect (CPE; Wenzel et al. 2020).  It just happened to also match the low density separation idea!  That's the key insight, which draws a surprising and non-obvious connection between two radically different phenomena: cold posteriors in Bayesian neural networks and semi-supervised learning.
>
> > How to determine a labelled/unlabelled observation is curated or not, beyond common data sets such as CIFAR10?
>
> Whenever we use a dataset, we should always understand how that dataset was created, perhaps by reading the documentation, or by asking the people who created the dataset.  This is true of very common dataset such as CIFAR-10, but is perhaps even more true with less common, and less well-understood datasets.  Part of this process will of course include understanding whether the dataset was curated.
>
> > Unlike penalty term which are artificial designed to incorporate prior knowledge, likelihood should reflect data generation rule and tuning-free.
>
> Do remember that the likelihood for e.g. neural networks, $P(y| X, w)$ has many, many tunable parameters: the neural network weights, $w$.  This likelihood may also include tunable hyperparameters, such as choice of leak in a leaky relu nonlinearity, or the choice of softmax temperature.  The choice of $S$ is conceptually no different.  Indeed, just as we could in principle learn these hyperparameters, we could in principle learn $S$.  However, learning $S$ only works when we have access to the noconsensus observations that were rejected during the curation process.  As we obviously do not usually have access to these rejected datapoints, it usually makes most sense to treat $S$ as a hyperparameter.
>
> > One experiment shows the improvement when unlabelled curated data is added, and the performance drop with more uncurated data. But this is not a convincing justification of the main of idea of the paper. The theory shows that log likelihood is lower bounded by the entropy or pseudo labelling loss under the generative model of curated data. It does not say what happens when the data is uncurated. To establish empirical justification, one also needs to show that the performance will increase with added uncurated data if likelihood (7) is used.
>
> We establish that the main idea works in Table 1, Fig. 3 and Fig. 4 (green + red lines).
>
> The issue appears to be that the blue and orange lines in Fig. 4 are not --- on their own --- a "convincing justification of the main idea of the paper".  They're not supposed to be.  That happens in Table 1, Fig. 3 and Fig. 4 (green + red lines).  They're just a simple sanity check, which turns out as expected.
>
> > The paper claims it is a Bayesian approach, but it never mention anything about prior setting. For all the results displayed, are they based on posterior MAP or posterior mean?
>
> We did also forget to specify the prior variance of the weights, an oversight that has now been fixed.  Specifically, we used independent Gaussian priors with variance 1/fan-in in Sec. 4.1 and variance $10^{-3}$ in Sec. 4.2.  All the Bayesian results are all based on the usual Bayesian model average (i.e. the posterior mean probability).
>
> > Figure 3, even when there is no unlabelled data, the test accuracy is different w.r.t. the choice of S. S=0 yields the best test accuracy. Please comment on this
>
> The lines for different values of S are not directly comparable, as they use different data. Specifically, each line uses data curated with that value for S, and as we increase S we curate more aggressively. Further, more aggressive curation will improve performance as it eliminates ambiguous or difficult-to-classify points. We have added a discussion of this point to the caption.
>
> > What is the test likelihood (Figure 3/4)?
>
> The test likelihood is just $P(Y_{\rm test}| X_{\rm test}, Y_{\rm train} X_{\rm train})$, (i.e.\ the likelihood for test points).  In practice, we use the Bayesian model avearge,
>
> $P(Y_{\rm test}| X_{\rm test}, Y_{\rm train}, X_{\rm train}) = \int dw P(Y_{\rm test}| X_{\rm test}, w) P(w| Y_{\rm train}, X_{\rm train})$

---

### Official Review · Reviewer_y8jM · 2022-10-28

**Confidence:** 4
**Correctness:** 3
**Technical Novelty And Significance:** 2
**Empirical Novelty And Significance:** 2
**Recommendation:** 5

**Clarity, Quality, Novelty And Reproducibility:**

Overall, the structure of the paper is organized; however, certain parts are difficult to follow. Data curation and principled likelihood estimates are two important concepts of this work. Although these two concepts have dependency, they have been used (the spread) over many different places in the paper which seems to be redundant. The readability of the paper can be improved if such redundancy can be simplified. The experiment section 4.1 is difficult to follow as it is missing some details such as the Gaussian parameters and how those were chosen, etc.

The paper contains some novelty: estimating the SSL lower bound in the context of data curation and experimenting and explaining the complexity as the uncertainty increases due to the increase of un-curated and unlabelled data.

As the code has been shared, it is expected the results can be reproduced although some additional details in section 4 could be helpful.


**Strength And Weaknesses:**

Strengths: The main idea of this paper is to derive a Bayesian formulation of the SSL setup by using labeled and unlabeled data. Using curated labeled data they have formulated/defined a lower bound of the solution. They also have proposed two likelihood estimates that are applicable to the curated labeled and unlabeled data and then combine them in a coherent fashion for model prediction.

The proposed methodology has been tested on CIFAR-10 and Galaxy Zoo datasets. Reported results are found to be better than FixMatch, especially when trained with less number of labeled data points (in the SSL mix).

Weaknesses: This work lacks a proper Bayesian formulation of the SSL problem which requires selection of some suitable priors and a proper estimation of the posteriors. Given that unlabeled data class labels are unknown it adds additional complexity to the modelling. There has been some prior work through the definition and usage of a null category (similar to the definition of the undefined class in this paper) such as [1, 2] which may help improve the Bayesian aspect of the formulation.

[1] https://papers.nips.cc/paper/2004/file/d3fad7d3634dbfb61018813546edbccb-Paper.pdf [2] http://www.bmva.org/bmvc/2011/proceedings/paper3/paper3.pdf)

**Summary Of The Paper:**

In this paper the authors have formulated Semi Supervised Learning (SSL) objective as a principled log-likelihood estimate in the context of a generative model of data curation. The distribution of data (in the classification context) has multiple or at least two levels of uncertainty: one is the distribution of the classes and the other is the labels themselves. The authors characterize the supervised setting as learning by using curated data labels (the lower bound of the SSL setting) and propose SSL as an extended case with curated plus non-curated and unlabelled data (coins the concept of undefined class while there is a disagreement on labels or no label available).

The proposed methodology has been tested on some synthetic, CIFAR, and Galaxy Zoo 2 datasets. Statistically significant improvements have been claimed over the FixMatch, a known technique in SSL.


**Summary Of The Review:**

I have gone through the paper more than once; overall, the idea sounds good and the results support some of the claims of this research. If we exclude certain sections, the paper is well written. The experiment section is missing some details and therefore found little difficult to follow.

The reported results are found to be interesting and provide us some insights on the difficulty of non curated and unlabelled data. I think this work has some value if some of the above mentioned limitations can be resolved/improved.

---

> ### Author Response · Authors · 2022-11-07
> **Response**
>
> > Weaknesses: This work lacks a proper Bayesian formulation of the SSL problem which requires selection of some suitable priors and a proper estimation of the posteriors.
>
> We're not quite sure what this means, as obtaining a proper Bayesian formulation of SSL was the whole point of the exercise!  Once you have our principled likelihoods, it is straightforward to combine the likelihood with standard BNN priors and inference methods, which is exactly what we did in the experiments!  We have added a very short section (Sec. 3.3) pointing this out.  We did also forget to specify the prior variance of the weights, an oversight that has now been fixed.  Specifically, we used independent Gaussian priors with variance $1/\text{fan-in}$ in Sec. 4.1 and variance $10^{-3}$ in Sec. 4.2.
>
> > There has been some prior work through the definition and usage of a null category (similar to the definition of the undefined class in this paper) such as [1, 2] which may help improve the Bayesian aspect of the formulation.
>
> This work is very useful if you have explicit OOD data, as the null category can be treated as the label for that data.  However, this approach doesn't work here because most SSL problems do not provide explicit null data.
>
> > The experiment section 4.1 is difficult to follow as it is missing some details such as the Gaussian parameters and how those were chosen, etc.
>
> We used independent Gaussian priors with variance of 1/fan-in for Sec. 4.1 and variance of $10^{-3}$ for Sec. 4.2.  We have also added more details, such as giving a reference to how the labels space for GZ2 was reduced, and the value of $S$ for the GZ2 experiment.
>
> > Data curation and principled likelihood estimates are two important concepts of this work. Although these two concepts have dependency, they have been used (the spread) over many different places in the paper which seems to be redundant. The readability of the paper can be improved if such redundancy can be simplified.
>
> We don't quite understand this comment.  We agree that it is a bit annoying that the theory is spread across the Background (Sec. 2) and Methods (Sec. 3), but that is necessary in order to delineate what theory is from the prior work and what theory is new.  Overall, we're happy to make any changes requested!

---

### Comment · Area_Chair_uird · 2022-11-14
**please discuss**

Hi reviewers. The authors have responded. Can I please ask you to engage with their response by posting further questions/comments here?

---

### Decision · Program_Chairs · 2023-01-20

**Decision:**

Accept: poster

**Justification For Why Not Higher Score:**

Some reviewers don't think the paper is ready in its current form.

**Justification For Why Not Lower Score:**

It might make sense to reject, see above.

**Metareview: Summary, Strengths And Weaknesses:**

The reviewers agreed that this paper makes a very useful contribution in terms of proposing a new likelihood for semi-supervised learning. However, the framing of the paper as a new _Bayesian_ approach to semi-supervised learning is not convincing: more methodological development and motivation, plus corresponding experiments would be needed to justify this. My judgment is that the paper is still a very useful one: the title should be changed (i.e. remove the word Bayesian) and the framing should simply state that Bayesian extensions are possible, but emphasize the main contribution in terms of a new likelihood.

**Note From Pc:**

if the above contains the word "oral" or "spotlight" please see: "oral" presentation means -> notable-top-5% and "spotlight" means -> notable-top-25%. As stated in our emails, we are disassociating presentation type from AC recommendations

**Summary Of Ac-Reviewer Meeting:**

We had a long discussion about the merits of the paper, and agreed that either the paper needed more experiments justifying the Bayesian part, or that needed to be deemphasized significantly. Since all the reviewers agreed the paper had merit, I think it should be accepted.